# Clinical Impact of Systematic Assessment and Psychoeducation in Specialized Treatment of Adolescents with Severe Functional Somatic Disorders: Results from the AHEAD Study

**DOI:** 10.3390/children10071101

**Published:** 2023-06-22

**Authors:** Karen Hansen Kallesøe, Kaare Bro Wellnitz, Eva Ørnbøl, Charlotte Ulrikka Rask

**Affiliations:** 1Department of Child and Adolescent Psychiatry, Aarhus University Hospital, Palle Juul-Jensens Boulevard 175, 8200 Aarhus, Denmark; karkal@rm.dk; 2The Research Clinic for Functional Disorders and Psychosomatics, Aarhus University Hospital, Palle Juul Jensens Boulevard 11, 8200 Aarhus, Denmark; kaare.bro@rm.dk (K.B.W.); eva.oernboel@aarhus.rm.dk (E.Ø.); 3Department of Clinical Medicine, Aarhus University, Incuba/Skejby Building 2, Palle Juul-Jensens Boulevard 82, 8200 Aarhus, Denmark

**Keywords:** adolescents, assessment, functional somatic disorders, functional somatic syndromes, psychoeducation

## Abstract

Functional somatic disorders (FSD), characterized by persistent and disabling physical symptoms, are common in adolescents. Diagnostic uncertainty and insufficient illness explanations are proposed perpetuating factors that may constitute barriers for treatment engagement. This study describes the impact of manualized assessment and psychoeducation on diagnostic certainty and various clinical outcomes in adolescents with multi-system FSD. Ninety-one adolescents (15–19 years) received systematic assessment (4 h) and a subsequent psychiatric consultation (1.5 h). Clinical characteristics included self-reported physical health, symptom severity, illness perception, illness-related behavior, and psychological flexibility assessed before and approximately two months after assessment, prior to specialized treatment. Data were analyzed using *t*-tests. Immediately following assessment, 71 (80.7%) adolescents out of 88 reported a higher diagnostic certainty and 74 (84.1%) reported that attending assessment gave them positive expectations for future treatment. A clinically relevant improvement of physical health was not observed at two months but considerable reductions were seen in symptom severity, illness worry, negative illness perceptions, illness-related limiting behavior, and psychological inflexibility. The results emphasize that systematic assessment and psychoeducation are important in their own right in the specialized treatment of adolescents with severe FSD.

## 1. Introduction

Functional somatic disorders (FSD), defined as disorders with distinct patterns of impairing physical symptoms with no clear medical explanation [1], are increasingly common in adolescents with current prevalence estimates of 3–10% [2,3,4,5]. The etiology of FSD is best understood within the bio-psycho-social model with often complex interacting biological and psychosocial factors for symptom development and maintenance [6]. Proposed maintaining factors include negative illness perceptions (e.g., low understanding of the disorder and expectations of long-term symptom duration) and maladaptive illness behavior (e.g., avoidance and ‘all or nothing’ behavior with a cycle of overdoing and excessively resting) [6,7,8]. Despite distress and high healthcare use [9,10,11], young patients and their parents are often left without clear explanations, diagnostic labels, or treatment advice when seeking medical care [12,13].

Several studies on youth with FSD have highlighted that the lack of tangible explanations for FSD may increase uncertainty in young patients and parents, which in turn causes distress, mistrust in the diagnostic label of FSD, and diminished engagement with available treatment options [12,14,15,16,17]. A qualitative study on adolescents with FSD and their parents concluded that there was a need for improved communication with clinicians, with extra focus on the discussion of results of medical investigations, especially negative findings with a lack of well-defined organic origins of symptoms [17]. In recent papers, the need for thorough assessment as a first step in the management of FSD, both in adults and youth, has been highlighted [6,13,18,19]. In adult patients with multi-system FSD, the experience and outcome of systematic assessment, psychoeducation, and follow-up consultation has been evaluated [20]. The study showed that such a systematic set-up was associated with clinically relevant improvements on symptom severity, illness worry, illness perceptions and behaviors, and positive expectations for treatment and future outcomes [20]. Positive effects of assessment and psychoeducation have also been reported in a study on children [21,22].

A potential barrier to the provision of clear and evidence-based psychoeducation is the use of various diagnostic labels for FSD. The heterogeneous symptom presentations of FSD, with various primary symptoms (e.g., fatigue, abdominal symptoms, or musculoskeletal symptoms) often influences the primary point of contact in the health care system (e.g., different sub-specialties in the pediatric setting or child and adolescent psychiatry). This will often influence the diagnostic categories used (e.g., different functional somatic syndromes, somatic symptom disorder, or functional neurological disorder). However, a large overlap has been shown between different diagnostic categories [23,24]. Based on this, a unifying classification system of FSD has been proposed, depending on the number of symptoms and involved symptom clusters (i.e., single-symptom, single-system, and multi-system) [1], recognizing the often more severely affected patients with multi-organ symptomatology [25,26].

The present study was part of a randomized controlled trial (RCT) testing a group-based intervention (Acceptance and Commitment Therapy for Health in Adolescents (AHEAD)) for adolescents with multi-system FSD [27,28]. The objective of the current study was to evaluate (1) how adolescents presenting with multi-organ symptomatology experienced systematic assessment (e.g., diagnostic certainty and outlook on illness course) and (2) whether systematic assessment and manualized psychoeducation would have a positive impact on self-perceived physical health, symptom severity, illness worry, and potential maladaptive illness perceptions and behaviors prior to further specialized treatment according to randomization in the overall trial.

## 2. Materials and Methods

### 2.1. Design

Selection for the study ran from January 2015 to December 2018. Patients were referred from general practitioners, medical specialists or hospital departments to a tertiary care clinic with special knowledge on assessment and treatment of multi-system FSD. Referrals were screened for eligibility and all potentially eligible adolescents were invited for assessment. Psychoeducation was provided immediately following assessment and, if consenting to study participation with no further time needed for consideration, the randomization ended the assessment day (see Figure 1). Patients were randomized to either AHEAD or enhanced usual care (EUC) with a personalized treatment plan for the general practitioner. A psychiatric consultation focusing on further psychoeducation and health-promoting strategies was scheduled approximately 2 weeks after the assessment. To ensure continuation, the same physician who performed the assessment both provided the subsequent psychoeducation and the following psychiatric consultation.

The study was conducted in accordance with the Declaration of Helsinki and the RCT was approved by The Danish Data Protection Agency (no. 1-16-02-290-14) and the Committee of Health Research Ethics of Central Denmark Region (no. 1-10-72-181-14). Trial registration before commencement: ClinicalTrials.gov NCT02346071.

### 2.2. Participants

Eligibility criteria for study participation were: age 15–19 years, fulfilment of multi-system FSD operationalized as multi-organ Bodily Distress Syndrome (BDS) [24] of at least one year’s duration, and clinician-rated moderate to severe impairment based on distress and impairment. Exclusion criteria were acute psychiatric disorder requiring other treatment, a lifetime diagnosis of psychosis, serious cognitive deficits, developmental disorders, substance abuse or pregnancy. All adolescents and parents (if age < 18) gave oral and written informed consent for inclusion before participation. 

Ninety-one adolescents were included in the study and attended both assessment and the following psychiatric consultation. Ninety percent of the included patients were female and had a mean symptom duration of 4 years. Forty-four percent had a present psychiatric comorbidity (i.e., anxiety and depressive disorders, or attention deficit disorder). Parents reported a lifetime history of FSD (35.2%), psychiatric disorders (44.4%), or substance abuse (12.4%). See Table 1 and RCT article for further details [28].

### 2.3. Assessment

Assessment was regarded as a pivotal part of the intervention, with five main foci, i.e., (1) systematic assessment of physical symptoms and potential psychiatric comorbidities, (2) creation of a chronological overview of health care contacts and social events, (3) clinical/neurological examination, (4) provision of a clear diagnosis, and (5) psychoeducation regarding FSD. See Figure 1 for overview of assessment and psychoeducation. The model for assessment was derived from a model developed and tested in an adult population [29] and was adapted for adolescents, e.g., with a larger degree of parental involvement and additional focus on assessment of potential underlying neurodevelopmental disorders. Prior to the assessment, the clinician reviewed all previous medical records, to ensure that the adolescent had been thoroughly medically examined according to symptom presentation, but also to assist the patient and parents in the creation of the chronological overview at the assessment.

#### 2.3.1. Clinical Interview

The adolescent and parents participated in a clinical interview focusing on childhood development from pregnancy/birth to present (see Figure 1). Both parents were encouraged to participate in the interview. To facilitate the possibility of linking life stressors (e.g., bullying, accidents, death of close relatives, parental divorce) with the development of physical symptoms a chronological overview of lifetime health care contacts and social events was made. The overview was made on a blackboard with previous symptoms, examinations, diagnoses and treatments on the left side of a timeline and important social events (both positive and challenging) on the right side (see Appendix A). 

#### 2.3.2. Assessment of Physical Symptoms and Potential Comorbidities

To ensure a systematic assessment of all physical symptoms and potential psychiatric comorbidities, the semi-structured diagnostic interview ‘Schedules for Clinical Assessment in Neuropsychiatry (SCAN)’ was used [30]. SCAN has a detailed section on functional somatic symptoms and also includes sections for screening and in-depth evaluation of general psychopathology. In addition, specific sections from the Development and Well-being Assessment (DAWBA) focusing on Attention Deficit Hyperactivity Disorder (ADHD), autism, and conduct disorder were used to screen for specific child psychiatry disorders not covered by the SCAN [31]. Assessment of potential underlying neurodevelopmental disorders is of importance as these disorders occur with higher prevalence in youth with FSD [32,33]. The physicians performing the assessment were trained in child and adolescent psychiatry, psychiatry, or community medicine, and also certified in conducting the SCAN interview.

#### 2.3.3. Clinical/Neurological Examination

Even though most patients had been extensively examined by their general practitioner or at somatic departments, a clinical/neurological examination was performed. It was prioritized in order to discover and address potential kinesiophobia and positive signs of FSD (e.g., paresthesia not following dermatomes) and to make an evaluation of how the patient handled a physical examination despite having many symptoms. The information from the clinical examination (e.g., potential positive signs of FSD, kinesiophobia) was further used for psychoeducation when addressing illness perception and behavior.

#### 2.3.4. Diagnosis

After finalizing the overall assessment, the clinician evaluated whether the adolescent fulfilled the diagnostic criteria of multisystem FSD (conceptualized as multi-organ BDS as described above). If so, the clinician explained in detail about the diagnosis and how this fit with the symptoms experienced by the adolescent. Furthermore, the adolescent and parents were ensured of the negative findings of the physical examination and of previous medical examinations.

### 2.4. Psychoeducation

Our intention was to provide the adolescents and parents with an evidence-based understanding of FSD and to facilitate a nuanced understanding of their illness. The psychoeducation consisted of (1) a bio-psycho-social explanatory model regarding the development and maintenance of bodily distress as the central feature of FSD, (2) a simple bodily focused explanation for symptom production and perception, and (3) perpetuating impact of maladaptive illness perceptions and behaviors.

The generic bio-psycho-social model regarding predisposing, precipitating and perpetuating factors was drawn on a blackboard and specific factors (e.g., important social events) mentioned by the adolescent and/or parents during assessment were incorporated in this model. The adolescent and parents were encouraged to openly express their reflections and understanding in the process in order to facilitate a discussion of their view, potentially shedding light on a purely biomedical understanding of symptom origin or different illness perceptions within the family. This overall discussion allowed the physician to clarify or address potential misunderstandings, e.g., of previous medical results or misconceptions of symptom development and the human body in general.Next step was a simple symptom explanation based on a model describing the presence of impairing symptoms as a combination of increased symptom production (arousal/‘stress’) and increased symptom perception (‘defect filter’). An outline of two persons was drawn on the blackboard, one person with FSD next to a person without FSD and the differences in filter and arousal. This model represented an evident simplification of the complex processes known to cause bodily distress and was therefore a clinical presentation of various physical symptoms corresponding to FSD. However, it provided a common language within the family and also a clearer understanding of why interventions (e.g., psychological or physiotherapy) may have a positive impact on the physical symptoms by targeting arousal and/or perception.Lastly, the specific illness-related behavior, ‘all-or-nothing’, was addressed and the inexpedient strategy of limiting or overdoing things. The adolescents were encouraged to aim for an activity level that was realistic without doing too little or too much. This was explained through an adapted ‘zone of proximal development’ model with three zones, i.e., comfort zone, development zone and overload zone. Illness perceptions were indirectly addressed throughout the assessment and psychoeducation, e.g., by broadening the perspective on symptom development and giving hope for symptom improvement through treatment.

For a detailed description of the bio-psycho-social model and introduction to common illness related behavior (all-or-nothing) see Additional File 1 in Kallesøe et al. [34]. Randomization was done at the end of the assessment day. Specific study information regarding the interventions was given prior to consent and randomization. The whole procedure including information and randomization took approximately 15 min.

### 2.5. Psychiatric Consultation 

The psychiatric consultation occurring approx. 2 weeks after the assessment focused on health-promoting strategies, i.e., sleep, diet, exercise, social network, and positive activities. A detailed history of each element was made to identify important elements for improvement. Current burdens were also disclosed. All elements (health-promoting strategies and burdens) were integrated elements of the ‘stress-resource fraction’ where stress was described as a low level of resources combined with high level of burdens while intervention for lowering stress was to improve resources and lower burdens (see Figure 2). After a thorough assessment of resources and burdens, the adolescent had to identify two specific elements to work on, e.g., to focus on improvement of sleep and graded exercise. Parents and/or other close relatives participated in the psychiatric consultation to help uncover resources and burdens but also to support the adolescent in the work elements they chose. 

### 2.6. Measures

#### 2.6.1. Evaluation of Assessment

Eight questions regarding the adolescents’ impression of the assessment, their certainty regarding their disorder, and their expectations for treatment were distributed the day following the assessment via a link in an email. The questions were answered before the follow-up psychiatric consultation. All eight questions can be seen in Section 3, Results.

#### 2.6.2. Other Outcomes

For evaluation of potential impact of systematic assessment, manualized psychoeducation and health promoting strategies the questionnaires chosen for the overall RCT-design were used. The questionnaires were distributed at baseline (before assessment) and approximately two months after assessment prior to engagement in specialized treatment.

Physical health (primary outcome in the RCT) was measured with an aggregate score deriving from the SF-36 subscales PF (physical functioning, 10 items), BP (bodily pain, 2 items) and VT (vitality, 4 items) shown to be sensitive to change in key areas affected in adults with FSD [29,35,36]. Scores range from 15–65 with higher scores indicating better physical health. A change of 4 and above may be regarded as a clinically relevant change and 8 and above as a marked improvement [37,38]. Danish sex and age-specific norm data are available from age 16 and up [39]. Internal consistency for the subscales measuring physical health were acceptable to good (Cronbach’s alpha PF 0.88, BP 0.80 and VT 0.72).

Symptom severity was measured with the somatization subscale of the Symptom Checklist Revised (12 items, 5-point scale, score range 0–4) [40] with higher scores indicating higher symptom severity. Internal consistency was good (0.83).

Illness worry was measured by Whiteley-6-R [41] (6 items, 5-point scale, score range 0–4), a validated modified version of the Whiteley Index. Higher scores indicate more severe symptoms of illness worry. Internal consistency was good (0.90). 

Illness perception was measured by the Brief Illness Perceptions Questionnaire (BIPQ) [42] (8 items, 11-point scale, score range 0–80). A higher score reflects a more threatening view of the illness. A review and meta-analysis of the B-IPQ has shown good psychometric properties across a range of populations and age-groups (8 to over 80) and has demonstrated sensitivity to change after intervention in randomised trials [42]. Internal consistency was acceptable (Cronbach’s alpha 0.65). 

Illness-related behavior was measured by two subscales of the Behavioural Responses to Illness Questionnaire (BRIQ) [43]: (1) all-or-nothing behavior (6 items, 5-point scale, score range 6–30) and (2) limiting behavior (excessive rest) (7 items, 5-point scale, score range 7–35) with higher scores indicating a higher degree of maladaptive illness-related behavior. Internal consistency was acceptable (Cronbach’s alpha for All-or-nothing 0.77 and Limiting 0.80). 

Psychological inflexibility was measured by two questionnaires. (1) The Avoidance and Fusion Questionnaire Youth (AFQ-Y8) (8 items, 5-point scale, score range 7–35 [44]. Recent studies show that AFQ-Y8 is a reliable measure of psychological inflexibility in children and adolescents [45,46]. Internal consistency was good (Cronbach’s alpha 0.90). (2) The Psychological Inflexibility in Pain (PIPS-12) consisting of two subscales: (a) Avoidance (8 items, 7-point scale, score range 8–56) and (b) Cognitive fusion (4 items, score range 4–28) [47]. Higher scores indicate a higher degree of psychological inflexibility (i.e., as a single construct or by avoidance and cognitive fusion, respectively). PIPS-12 has been validated age 17 and up [47]. Internal consistency was good for the Avoidance subscale (Cronbach’s alpha 0.90) and acceptable for the Cognitive fusion subscale (Cronbach’s alpha 0.72). The concept of psychological flexibility pertains specifically to Acceptance and Commitment Therapy (ACT), and describes the ability to stay in contact with the present moment regardless of unpleasant thoughts, feelings, and bodily sensations, while choosing one’s behaviors based on the situation and personal values [48].

### 2.7. Analysis

Descriptive statistics were used to characterize the sample at baseline. Data were summarized as either mean and standard deviation (SD) or as count and percentage, depending on variables. Patients’ evaluations of the clinical assessment were presented as percentage. Clinical outcome data were analyzed using paired *t*-tests from baseline to two months after assessment. The standardized response mean (SRM) is calculated as an effect size index. In order to investigate whether the randomization had any influence, mixed models with random intercept, and intervention, time, and their interaction as independent variables were estimated for each of the clinical outcomes. Due to randomization, the main effect of intervention was fixed to zero to reduce bias from potentially different baseline values [49]. Assumptions of normality and homoscedasticity of residuals and random effects were visually inspected using scatter-, QQ-, and box-plots. Analyses were performed using Stata version 17.0 for Windows.

## 3. Results

### 3.1. Evaluation of Assessment

Eighty-eight patients (96.7%) answered the questionnaire regarding the experience of assessment. The majority of patients reported that it was a positive experience attending assessment, that they felt less uncertain of what was wrong with them, and that it had given them a better understanding of their disorder (see Figure 3). One fourth of patients reported that they had improved somewhat and almost half of patients reported that it had given them specific ideas on how to get better. Most patients reported positive expectations for future treatment and two thirds reported that it made the future look brighter. Twenty-three patients (26.1%) agreed or partly agreed that assessment had not changed anything, 7 of which had been randomly assigned to AHEAD and 16 of which had been randomly assigned to EUC. Experience of assessment divided by randomization groups can be seen in Appendix A.

### 3.2. Preliminary Change after Assessment and Psychoeducation

Time between assessment and evaluation at T1 prior to start of specialized treatment was a median of 62 days, (IQR 28-103). Assessment and psychiatric consultation did not in itself cause a clinically relevant improvement of physical health (see Table 2). With regard to impact on secondary outcomes and treatment targets, respectively, there was a decline in symptom severity and illness worry as well as in negative illness perception, illness-related limiting behavior, and psychological inflexibility on one scale (both sub-scales of PIPS-12). At T1, there were no observed differences in improvements between the AHEAD and EUC groups in all outcomes (see interaction effects Table 2). Mean and SD at T0 and T1 for both randomized groups can be seen in Appendix A.

## 4. Discussion

The present study has shown that a thorough clinical assessment and psychoeducation was associated with diagnostic certainty, positive outlook on future treatment and a general hope for a better future for the young patients. Furthermore, the combination of assessment, psychoeducation, and health-promoting strategies was associated with an improvement in symptom severity, illness worry, illness perception, illness-related limiting behavior, and psychological flexibility two months after assessment. 

The negative consequences of not providing a clear diagnosis has been highlighted in previous pediatric studies. A qualitative study in adolescents with physical symptoms without organic pathology addressed that a lack of diagnosis and medical explanation for symptoms is difficult for patients and their parents to accept [17]. Moulin et al. further observed that diagnostic uncertainty fueled uncertainty of how to handle symptoms in everyday life but also had emotional consequences where parents sometimes wished for more serious pathology just to have a diagnosis that could alleviate the disbelief from their social circles [17]. A paper presenting clinical vignettes illustrated a lack of diagnosis and dismissive and misguided attitudes of healthcare professionals leave young patients and their parents with a sense of not being taken seriously [13]. The experiences of adults also underscore that medical reassurance is not sufficient when targeting diagnostic uncertainty [50]. When young patients and their parents are not met with individualized bio-psycho-social explanations taking all life aspects into account, including, e.g., infections, social life-stressors, and psychological aspects, the young patients and their parents may find their own justification for the symptoms often focusing on biomedical/physical factors [8,17]. The tendency to seek a biomedical explanation may stand in the way of relevant treatment options, including psychological interventions. Hence, using a bio-psycho-social framework in the assessment of a young patient with FSD in a mindful way, to explore potential complex interactions is essential before more specialized treatment with psychotherapy [13,51]. 

The vast majority of adolescents in this study reported that attending assessment was a positive experience indicating that the bio-psycho-social model can be successfully implemented, balancing all elements of the model including the sharing of psychological factors (e.g., personal, sensitive information) without making the adolescents feel dismissed by the physician. This is a valuable finding, as research has shown that there is a risk of feeling dismissed when the explanation is not approached respectfully [12,13,52]. Furthermore, more than four out of five adolescents reported that attending assessment gave them positive expectations for future treatment and many also agreed that it made the future look brighter in general. In the adult literature, it has been shown that positive treatment expectations are associated with a better overall outcome of treatment [53,54,55]. The impact of treatment expectations has not been thoroughly examined in youth but a study in children with chronic pain did show that a higher degree of readiness to self-manage pain was associated with greater improvements from pre- to post-treatment and the use of more adaptive coping strategies [56]. As the systematic assessment and psychoeducation in the present study succeeded in creating positive expectations and general hope, this may be an important step towards better outcomes, as suggested by the adult literature. 

Two months after assessment, the overall intervention, including psychoeducation and personalized health-promoting strategies, was associated with an early reduction in symptom severity, illness worry, negative illness perceptions, limiting illness-related behavior, and psychological inflexibility. In a study on FSD in adults, an early treatment response was predictive of a better treatment outcome, especially regarding illness worry [57]. In comparison, early predictors of a positive treatment response have not been thoroughly investigated in youth. However, specific factors that may be important to address in treatment have been identified in mediation studies across different single-system FSDs in children and adolescents [58]. These include specific behaviors and perceptions, e.g., improvement of avoidance/limiting behavior [59,60], and catastrophic cognitions [61] in gastrointestinal FSD and pain-impairment beliefs in children with chronic pain [62]. It could therefore be suggested that these factors should be considered early on in psychoeducation, as done in the present study. The early improvements seen in the present study may serve as a first step towards a broader and more adaptive understanding of and approach to the disorder. The positive impact of psychoeducation in itself has been shown in a pediatric RCT study on the effect of cognitive behavioral therapy (CBT) for chronic fatigue syndrome by Chalder et al. [22], although CBT displayed important advantages at long-term follow-up [21]. This indicates that psychoeducation may serve as a relevant minimal first step intervention, though more intensive treatment will be necessary for patients with more severe clinical presentations.

The present study has several limitations. First, the study design with a lack of a control group for comparison limits the overall conclusions regarding the specific impact of the described assessment, psychoeducation, and health-promoting strategies. Second, this is a study embedded in the AHEAD trial and, therefore, not optimally designed to evaluate the specific effects of assessment, psychoeducation, and health-promoting strategies. The observed changes from baseline to approximately two months after assessment may therefore be due to spontaneous improvements. However, as the adolescents had been ill for a mean of 3.9 years with multi-system involvement, spontaneous improvements are expected to be low. Third, randomization was done right after assessment, prior to evaluation of the effect of assessment and psychoeducation, potentially influencing outcome with different incentives to work with the health-promoting strategies. However, the interaction effects did not suggest that randomization group influenced change. Fourth, the time period between T0 and T1 differed in length due to the commencement of group treatment defining the T1 measurement, with some patients having a short time between the two measurement points, which may have influenced the potential for improvement. Fifth, the design did not leave room for an evaluation of the impact of assessment and psychoeducation as a standalone intervention, and it is therefore not possible to evaluate the long-term effects on relevant outcomes such as symptom severity and healthcare use. Lastly, despite the importance of parental factors, we did not evaluate the parents’ experience of assessment, which may differ from the experience of the adolescents, potentially influencing the course of symptoms. 

## 5. Conclusions

The present study has shown that systematic assessment and psychoeducation were associated with diagnostic certainty, positive treatment expectations, and a brighter outlook on the future for the young patients diagnosed with FSD. Improvements were observed on important clinical outcomes including symptom severity, illness worry, illness perception, illness-related limiting behavior, and psychological flexibility prior to specialized treatment. The results underscore the importance and potential positive implications of systematic assessment and psychoeducation. Future research should investigate systematic assessment and psychoeducation in a randomized design to evaluate long-term effects and the relevant amount needed.

## Figures and Tables

**Figure 1 children-10-01101-f001:**
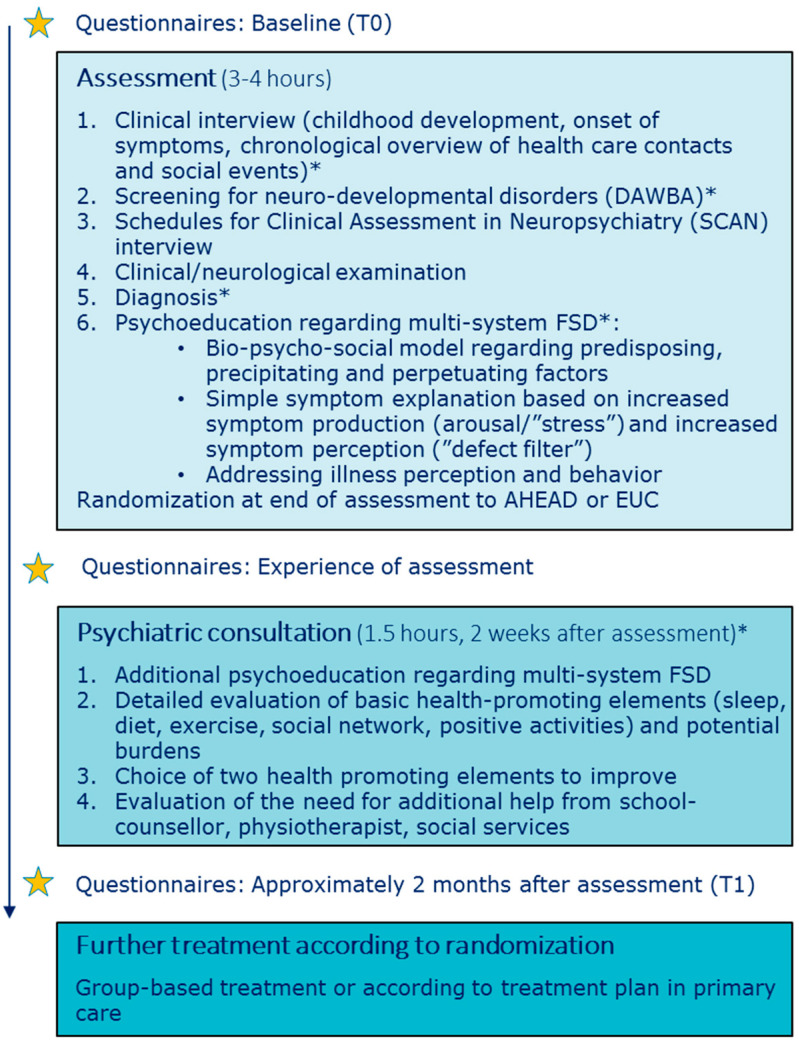
Chronological overview of assessment and psychoeducation. * Participation of parents/close relatives AHEAD: Acceptance and Commitment Therapy for Health in Adolescents, DAWBA: Development and Well-Being Assessment, EUC: Enhanced Usual Care, FSD: Functional Somatic Disorders.

**Figure 2 children-10-01101-f002:**
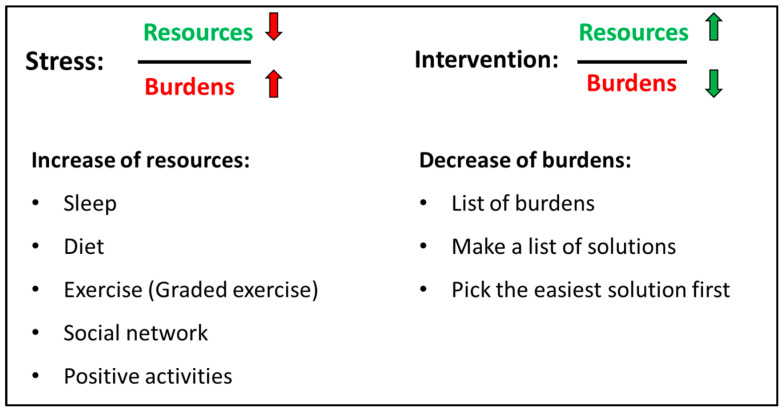
The stress-resource fraction [34]. Stress is explained as low level of resources combined with high level of burdens (red arrows). The intervention for lowering stress is to improve resources (e.g., by improving sleep, exercise or social network) and lower specific burdens identified by the patient and parents (green arrows).

**Figure 3 children-10-01101-f003:**
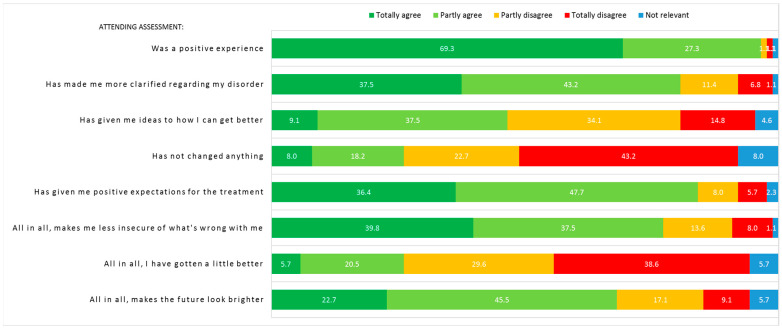
Experience of assessment (*n* = 88). Numbers in figure represent percentage. Be aware of item 4: “Has not changed anything”, where the orange (partly disagree) and red (totally disagree) answers counterintuitively equals the patients with an impression of change after assessment.

**Table 1 children-10-01101-t001:** Patient characteristics at baseline.

	N = 91
**Sex**, female: *n* (%)	82 (90.1)
**Age at inclusion**, years ^1^	17.9 (1.5)
**Symptom duration**, years ^1^	3.9 (2.1)
**Psychiatric comorbidity**, *n* (%) ^2^	
1. Current anxiety disorder	30 (33.0)
2. Current depressive disorder	22 (24.2)
3. Attention deficit disorder	3 (3.3)
4. Any	40 (44.0)
**Physical Health**, SF-36 Aggregate Score (15–65) ^1, 3^	36.8 (6.9)
**Symptom score**, SCL-somatization (0–4) ^1^	1.9 (0.8)
**Mental component score** (MCS) ^1, 4^	35.7 (14.2)
**Illness worry** (0–4) ^1, 5^	1.7 (1.1)
**Clinician rated impairment in daily life**, *n* (%)	
Moderate	25 (27.5)
Severe	66 (72.5)
**School or work attendance**, *n* (%) ^6^	
Normal conditions	2 (2.2)
High degree of absence, special conditions	75 (82.4)
No school or work attendance	14 (15.4)
**Parental cohabitation** (living together: *n* (%)) ^6^	56 (61.5)
**Father’s highest level of education** (n (%)) ^6^	
Short (high school or below)	27 (29.7)
Medium (vocational, bachelor or equivalent)	39 (42.9)
Higher (master or equivalent)	16 (17.6)
Absent	9 (9.9)
**Mother’s highest level of education** (*n* (%)) ^6^	
Short (high school or below)	26 (28.6)
Medium (vocational, bachelor or equivalent)	52 (57.1)
Higher (master or equivalent)	8 (8.8)
Absent	5 (5.5)
**Parental lifetime history of**: (*n* (%)) ^6^	
Functional somatic disorder	32 (35.2)
Psychiatric disorder	40 (44.4)
Substance abuse	11 (12.4)

^1^: Mean (SD); ^2^: present diagnoses evaluated by clinician at assessment; ^3^: aggregate score of three SF-36 subscales, i.e., physical functioning, bodily pain, and vitality (range 15–65); ^4^: SF-36 mental component Score; ^5^: Whiteley-6-R; ^6^: anamnestic information.

**Table 2 children-10-01101-t002:** Unadjusted change scores and interaction effects from baseline (T0) to two months after assessment (T1).

		*t*-Test				Mixed Analysis		
		Difference	95% CI	*p*-Value	SRM	Interaction Effect	95% CI	Missing n
**Primary outcome**								
Physical health	SF-36 (15–65)	0.23	(−0.95; 1.41)	0.701	0.04	−1.46	(−3.59; 0.68)	6
**Secondary outcomes**								
Symptom severity	SCL-som (0–4)	−0.15	(−0.27; −0.03)	0.017	0.26	0.02	(−0.20; 0.25)	2
Illness worry	Whiteley-6-R (0–4)	−0.63	(−0.79; −0.47)	<0.001	0.86	0.21	(−0.08; 0.50)	5
Mental health	SF-36 MCS	0.77	(−1.72; 3.27)	0.539	0.07	0.39	(−4.11; 4.88)	6
**Treatment targets**								
Illness perception	B-IPQ (0–80)	−4.24	(−6.03; −2.44)	<0.001	0.50	2.65	(−0.44; 5.75)	2
Illness-related behavior	BRIQ-All or nothing (6–30)	−0.74	(−1.64; 0.17)	0.108	0.17	−0.23	(−1.80; 1.35)	3
	BRIQ-Limiting (7–35)	−1.49	(−2.41; −0.57)	0.002	0.34	0.28	(−1.37; 1.93)	3
Psychological inflexibility	AFQ-Y8 (0–32)	−0.55	(−1.52; 0.41)	0.258	0.12	1.61	(−0.19; 3.40)	6
	PIPS-Avoidance (8–56)	−2.42	(−4.03; −0.82)	0.003	0.33	−0.95	(−3.92; 2.01)	6
	PIPS-Fusion (4–28)	−0.99	(−1.90; −0.08)	0.034	0.23	0.77	(−0.82; 2.37)	6

Table 2 displays the unadjusted mean differences and their 95% confidence intervals from baseline (T0) to right before start of specialized treatment (T1), approx. two months apart, for all patients included in the RCT. A negative value means a decrease in the respective scores. For physical health and mental health, a positive value means an increase in self-reported physical and mental health. Interaction effects and their 95% confidence intervals from the mixed analysis are also displayed, to evaluate whether randomization influenced change. AFQ-Y8: Avoidance and Fusion Questionnaire Youth; B-IPQ: Illness Perception Questionnaire; BRIQ: Behavioural Response to Illness Questionnaire; CI: Confidence interval; MCS: Mental component summary; PIPS: Psychological Inflexibility in Pain questionnaire; SF-36: The Short Form (36) Health Survey; SCL-som: Symptom Checklist Revised somatization subscale; SRM: standardized response mean.

## Data Availability

The individual participant data that underlie the results reported in this article will be available for researchers who provide a methodologically sound proposal to achieve their aims. Proposals should be directed to the first author, karkal@rm.dk, to gain access. Data requestors will need to sign a data access agreement. The data are available immediately following publication and no specific end date has been set.

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
