# Peer review of "Clinical Impact of Systematic Assessment and Psychoeducation in Specialized Treatment of Adolescents with Severe Functional Somatic Disorders: Results from the AHEAD Study"

_children, 2023, doi:10.3390/children10071101_

Round 1

Reviewer 1 Report

Thank you for the opportunity to review this manuscript. It was well written and will be of interest to the journal readers. This research focuses on two areas of research infrequently addressed within the literature on adolescents who present with severe functional somatic disorders: (a) perceptions of diagnostic certainty and outlook on illness course and (b) the role of assessment and manualized psychoeducation in improving biopsychosocial functioning. Pertaining to the first research objective, more than 8 in 10 participating adolescents reported positive perceptions of the systematic assessment/psychoeducation approach to their diagnosis and prognosis. A lack of a comparison group undergoing assessment procedures “as usual” would certainly have helped to better understand the implications of findings. However, the findings around service satisfaction provide readers with a broader vision of assessment procedures that may improve the care of patients with severe somatic concerns, even without a control or comparison group. The second question focused on the impact of systematic assessment/psychoeducation on adolescent perceived behavioral health indicators. While no impact was seen on physical health which was disappointing to read, there were noted improvements on numerous psychological outcomes. Again, a comparison group would have helped to improve the interpretability of those findings. In addition, reliance on more than adolescent perceptions of behavioral health improvements would have further added to the meaningfulness of results. Despite those method limitations, the study findings are both original and relevant and help to address gaps identified in the literature. Numerous practical ideas for the assessment and treatment of patients presenting with severe somatic arise from the study findings. Examples include the importance of a comprehensive assessment approach and the potential benefits of psychoeducation/health promotion on improving psychological functioning (e.g., symptom severity, illness perception/worry, functional impairment, and psychological flexibility. Again, without a comparison or control group, the implications of study findings are scientifically limited, yet practically meaningful for behavioral professionals working with adolescents with severe somatic concerns. The authors are encouraged to bring additional methodological rigor to their future research endeavors to verify the generalizability of study findings with the study population. For example, using a standard care condition, a systematic assessment/psychoeducation condition, and a systematic assessment/psychoeducation/treatment condition would help to further isolate the specific impact of the conditions investigated within this study. Study findings are contextualized effectively within the relevant prior literature. I note that the tables, figures, and supplementary material all appear to support the efficiency and effectiveness of the study write-up. I especially appreciated their use of “stop light” colors to indicate positive/negative care perceptions. Overall, this was a very well-written manuscript. Despite a number of methodological limitations, study findings are relevant to a range of behavioral health professionals and further advance the diagnosis and treatment of adolescents presenting to care with severe and dysfunctional somatic symptoms. The authors are to be commended for their exceptional presentation of their study and the connections made to the current literature. The only minor edits I noted were in Figure 3 where the percentages represented seemed to include a comma instead of a decimal (period) in the pdf file that I reviewed, and within the references which appeared to be formatted differently than the journal formatting requirements

Author Response

Thank you so much for the positive review of our manuscript.

We very much agree on the limitations in study design with lack of control group and have therefore chosen to address this further in the limitations line 454-457.

Regarding the lack of improvement on physical health: The aggregate score of physical health consists of the subscales physical functioning, bodily pain and vitality from the SF-36. For some of the parameters we do not expect a fast improvement, as the adolescents have a mean symptom duration of almost 4 years. We also expect that they will need more treatment than psychoeducation and focus on health promoting strategies. However, we did see smaller improvements of physical health in the EUC condition from T0 to T1 but not in the AHEAD condition (see Table S1). It could be speculated whether the patients receiving the EUC condition worked more with the health promoting strategies than the patients waiting for the group-based intervention. These are just speculations and the design does not allow for us to answer this question as stated in the limitation section.

Regarding the use of comma instead of period in Figure 3 and supplementary Figure S2 this has now been corrected.

Reviewer 2 Report

Thank you for inviting me to review this article with the following objectives:

to evaluate: 1) how adolescents presenting with multi-organ symptomatology experienced systematic assessment (e.g. diagnostic certainty and outlook on illness course) and

2) whether systematic assessment and manualized psychoeducation would have a positive impact on self-perceived physical health, symptom severity, illness worry, and potential maladaptive illness perceptions and behaviors

The first objective was examined by using a questionnaire filled in by 88 patients. The second objective was examined through a package of questionnaires filled in at baseline and after two months, but before the patient started any intervention. I see no problem with the first objective and how it was examined, but with the second I miss a control group for comparison. Moreover, the relation to the RCT is confusingly described, and the following RCT is in fact only something that might have biased the results, but is actually not part of this study methodology. I think the current description is unclear. Therefore, I have suggestions for major revisions.

My comment in more detail follows below.

Abstract

Line 16; change evaluates to describes

Line 17, 19; clinical outcomes could be change to clinical characteristics

Line 18; Remove this “included in a randomized trial testing group-based therapy (AHEAD)”

Introduction

I think this is clear, and have no comments

Materials and methods

2.1 Design

Line 81-82; This should be rewritten so that focus is on assessment and manualized psychoeducation, and when they were conducted in relation to the randomization. Now it is hard to understand when they are performed. Both points of assessment should also be described, and their relation to randomization

2.6 Measures

These eight questions are crucial, if not written out in the text, there should be a reference to Figure 3

2.7 Analysis

Add effect sizes

Results

Since the sample description is presented as result there should be some research question about its characteristics, otherwise it should be presented under material.

In Table 2 I would like to see effect sizes.

Discussion

Line 336; delete “embedded in the AHEAD trial”

Line 337; I would prefer another wording, because you don´t know if ordinary assessment and information would have had the same impact on symptoms, maybe use “related” instead of “improved”

Line 343-344; delete this “A higher diagnostic certainty was obtained through systematic assessment, provision of a clear diagnosis (i.e. multi-system FSD) and manualized psychoeducation with inclusion of personal elements from the systematic assessment”

Line 418; or somewhere in the last section there should be some discussion about a limitation of not having a control group that had received standard assessment and information for comparison.

Conclusions

Line 320; delete “embedded in the AHEAD trial”, rewrite it. Assessment and psychoeducation seem to have a positive impact but we don´t know if this amount of assessment and education is needed. The wording should mirror that.

Reviewer 3 Report

Thank you for the opportunity to revise this paper. Despite the originality of the study, I cannot understand the objective of the study. I do not believe that the work is sufficiently yet mature to be published.

Major Points:

Abstract & Introduction

1.      Although the summary and introduction are written clearly, they are far from well conceptualized. The introduction does not provide a detailed definition of this disorder and does not even distinguish it from other similar diagnostic patterns. Moreover, the incidence rate suggested in the article seems too high for this specific disorder, as it is possibly linked to somatoform disorder in general according to the literature.

2.      There is no theoretical framework provided to explain the origin and maintenance of this disorder. This is a basic gap because without understanding the phenomenon, any treatment strategies seem to have no future.

3.      There are no sufficiently described efficacy studies reported that guarantee that Acceptance and Commitment Therapy influences this type of disorder.

4.      I also think that the efficacy of this type of intervention should be questioned in multisystemic patients, meaning those with significant variability.

5.      There are no predictions from the study.

Methods:

1.      There is no reference to any exclusion criteria related to medication use.

2.      I think that, in the case of children and adolescents, an informed consent should have been completed by parents or caregivers.

3.      The methods section is also disorganized in places and lacking in detail. Paragraph structure seems haphazard in places (in general, one-sentence paragraphs should be avoided).

4.       Neither in the evaluation section nor in the intervention is it mentioned whether it is done by the same professionals (Psychologists or Psychiatrists) or by different ones - I think this variable could be parasitic.

5.       Were the assessment instruments randomized or did they always follow the same order?

6.      None of the instruments (evaluation and results) have psychometric characteristics, how do we know if they are reliable?

7.      There is no section dedicated to the general procedure of the study

8.      The statistical procedure seem to me to be adequate.

Results

1.      They are clear and appear conveniently described.

Discussion:

1.The authors referred that improvements were observed on important clinical outcomes including symptom severity, illness worry, illness perception, limiting behaviours and psychological flexibility prior to specialized treatment. However, there were some baseline data that made me apprehensive. From what I understand, for example, the values of illness worry were 1.7 (on a scale of 0-4). These values are below half... are they not typical for this age?

2. How do you explain that there was no relevant improvement in physical health? it doesn't seem to me to be just explained by the design of the program.

3. Overall, I find the discussion poor because the introduction is also limited. I think the speech is very simplistic and lacks a theoretical depth that explains, from a cognitive point of view, how this disorder is processed and how we can intervene in it.

Round 2

Reviewer 2 Report

Thank you for addressing all points raised. I have no further suggestions

Reviewer 3 Report

The authors made an effort to make improvements to the paper. However, I have doubts as to its scientific relevance.